# Epigenetic Reprogramming in *Mice* and *Humans*: From Fertilization to Primordial Germ Cell Development

**DOI:** 10.3390/cells12141874

**Published:** 2023-07-17

**Authors:** Aditi Singh, Daniel A. Rappolee, Douglas M. Ruden

**Affiliations:** 1CS Mott Center, Department of Obstetrics and Gynecology, Wayne State University, Detroit, MI 48202, USA; hc2799@wayne.edu (A.S.); drapploe@med.wayne.edu (D.A.R.); 2Center for Molecular Medicine and Genetics, Wayne State University, Detroit, MI 48202, USA; 3Reproductive Stress Measurement, Mechanisms and Management, Corp., 135 Lake Shore Rd., Grosse Pointe Farms, MI 48236, USA; 4Institute of Environmental Health Sciences, Wayne State University, Detroit, MI 48202, USA; 5Department of Physiology, Wayne State University, Detroit, MI 48202, USA

**Keywords:** epigenetic reprogramming, developmental toxicity, embryo development

## Abstract

In this review, advances in the understanding of epigenetic reprogramming from fertilization to the development of primordial germline cells in a mouse and *human* embryo are discussed. To gain insights into the molecular underpinnings of various diseases, it is essential to comprehend the intricate interplay between genetic, epigenetic, and environmental factors during cellular reprogramming and embryonic differentiation. An increasing range of diseases, including cancer and developmental disorders, have been linked to alterations in DNA methylation and histone modifications. Global epigenetic reprogramming occurs in mammals at two stages: post-fertilization and during the development of primordial germ cells (PGC). Epigenetic reprogramming after fertilization involves rapid demethylation of the paternal genome mediated through active and passive DNA demethylation, and gradual demethylation in the maternal genome through passive DNA demethylation. The de novo DNA methyltransferase enzymes, *Dnmt3a* and *Dnmt3b*, restore DNA methylation beginning from the blastocyst stage until the formation of the gastrula, and DNA maintenance methyltransferase, *Dnmt1*, maintains methylation in the somatic cells. The PGC undergo a second round of global demethylation after allocation during the formative pluripotent stage before gastrulation, where the imprints and the methylation marks on the transposable elements known as retrotransposons, including long interspersed nuclear elements (LINE-1) and intracisternal A-particle (IAP) elements are demethylated as well. Finally, DNA methylation is restored in the PGC at the implantation stage including sex-specific imprints corresponding to the sex of the embryo. This review introduces a novel perspective by uncovering how toxicants and stress stimuli impact the critical period of allocation during formative pluripotency, potentially influencing both the quantity and quality of PGCs. Furthermore, the comprehensive comparison of epigenetic events between *mice* and *humans* breaks new ground, empowering researchers to make informed decisions regarding the suitability of mouse models for their experiments.

## 1. Epigenetic Reprogramming: Overview

The oocyte and sperm are highly specialized cells that contain nuclear genes which are differently modified and carry their own epigenetic marks such as DNA methylation, histone modifications, and chromatin organization, all of which regulate the expression of genes. The oocyte, being a fully differentiated cell, possesses a distinct DNA methylation pattern that sets it apart from sperm or somatic cells. In sperm cells, DNA methylation is evenly distributed throughout the genome, covering roughly 90% of it, except for certain CpG islands (CGIs) that evade methylation. Conversely, mature oocytes exhibit a global DNA methylation level of approximately 40% [1,2,3]. Upon fertilization, a process called zygotic genome activation (ZGA) occurs, which involves the erasure of many of the epigenetic marks on the DNA coming from maternal and paternal genomes, providing a clean slate [4]. The ZGA in mammals is completed by the two-cell embryo, and new RNA transcripts are made which helps in generating the trophectoderm and inner cell mass (ICM) lineage specification [5]. The paternal genome undergoes rapid, active DNA demethylation, which is the demethylation mediated by enzymes ten-eleven translocation methylcytosine dioxygenase 1 and 2 (*TET1* and *TET2*). However, the maternal genome appears to not utilize active demethylation, and only undergoes gradual, passive demethylation, which is caused by multiple rounds of replication of the DNA in the absence of DNA methylation maintenance machinery [6].

The process of DNA methylation, particularly at DNA CpG islands and shores, plays a fundamental role in gene regulation and genomic stability [7]. In addition to CpG methylation, emerging evidence highlights the presence of non-CpG methylation, including CpA, CpT, and CpC, which expands the complexity of DNA methylation patterns [8]. DNA methylation at CpG islands typically leads to transcriptional repression, while methylation at CpG shores, located in close proximity to islands, can also impact gene expression [9]. Several key enzymes are responsible for executing DNA methylation and demethylation processes in both *mice* and *humans*. In *mice* and *humans*, DNA methylation is predominantly mediated by DNA methyltransferases (DNMTs), including *DNMT1, DNMT3A*, and *DNMT3B*. These enzymes establish and maintain DNA methylation patterns during development and throughout life, along with the ten-eleven translocation (*TET*) family of enzymes that catalyze active DNA demethylation by iterative oxidation reactions. Understanding the dynamic interplay between DNA methylation and these enzymatic processes is crucial for unraveling the intricate regulatory mechanisms underlying gene expression and epigenetic modifications in health and disease.

By the morula stage, the DNA methylation pattern of most of the CpG islands in the embryo, which are primarily found in the proximal promoter regions of most genes and control gene expression, is essentially erased. Interestingly, stochastic mechanisms result in differential DNA methylation patterns in certain regions, including gene bodies, CpG islands, promoters, and enhancers, which are low in repetitive elements. These differentially methylated regions have been proposed as potential hotspots for transgenerational epigenetic inheritance, as they may contribute to the transmission of epigenetic information across generations. Notably, during global DNA demethylation processes in PGC, the evolutionarily young and frequently intact retrotransposons, such as SINE/variable number of tandem repeats/Alu (SVA) elements in *humans* and intracisternal-A particles in *mice*, exhibit differential DNA methylation patterns. This distinctive methylation status may facilitate their expression and transposition, highlighting their potential significance in developmental processes [10]. It should be noted that the germline employs different epigenetic, transcriptional, and posttranscriptional mechanisms to regulate retroelements/transposable elements. These regulatory processes serve to safeguard the germ line and uphold genomic integrity [11,12].

In the context of PGC development, the mouse Vasa homolog (*Mvh*) has been identified as a crucial factor. *Mvh* is involved in the specification of PGCs in *mice*, and males lacking *Mvh* due to knock-out mutations exhibit infertility [13]. In *Drosophila*, Vasa mutants were discovered in a female-sterile genetic screen and found to lack pole cells which later become germline stem cells [14]. Similar to *mice, humans* with a non-functional Vasa homology creates only male infertility [15]. Besides *Mvh*, several other PGC proteins in *mice* are essential for fertility, including *Stella/Dppa3, Blimp/Prdm1, Tcfap2c/AP2γ,* and *Prdm14*. The functions of these proteins will be discussed in detail later in this article.

The de novo DNA methylation blastoderm cells continues until the gastrula stage, and then cell-type-specific DNA methylation patterns in the differentiated cell types are maintained in the growing embryo. At the gastrula implantation stage, PGCs undergo a second round of active demethylation of the entire genome, including the imprinting control regions for retrotransposons, including LINE-1 and IAP elements, leading to the most demethylated cells in the mammalian life cycle. Preparation for sexual reproduction begins with erasing somatic signatures in the PGCs through an extensive reprogramming process, establishing the germ-cell specific transcription profiles and epigenetic signatures for the meiotic maturation and fertilization [16]. After complete erasure of the DNA methylation, de novo methylation of the PGCs is where new DNA methylation patterns are re-established, such as the imprinting control regions to reflect the sex of the embryo [17].

## 2. DNA Demethylation: First Wave after Fertilization

Upon fertilization, the *mouse* zygote genome undergoes massive reprogramming. Demethylation results from the absence of DNA methylation maintenance, carried out by *Dnmt1*, and rapid demethylation by the *Tet* enzymes. The male and female genomes are reprogrammed at different rates. It is known that the paternal genome undergoes rapid demethylation and that the demethylation in the maternal genome is comparatively slower. (Figure 1) [18]. The paternal demethylation is facilitated by *Tet3*, which oxidizes 5-methylcytosine (5mC) to 5-hydroxymethylcytosine (5hmC) by active DNA demethylation [19]. The Tet enzymes (*Tet1/2/3*) have catalytic domains harboring an essential metal-binding residue in their double-stranded β-helix fold [20]. Tet enzymes also require a cysteine-rich domain to function. *Tet1/3* bind to methylated CpG islands via their chromatin-binding CXXC domain [21]. *Tet2* lacks this CXXC chromatin domain and functions in conjunction with the *IDAX* protein, which has an independent CXXC-containing protein [22].

Maternal DNA undergoes demethylation passively and gradually, in comparison to the paternal genome. The maternal genome actively excludes *Tet3*, whereas the paternal genome has factors that bind *Tet3*. The *PGC7* gene codes for *Dppa3/Stella*, which is expressed during PGC allocation in peri-gastrulation embryos [23]. A study using single-cell RNA sequencing technique found that the genes *Stella* and *Esg1* have opposite patterns of expression during early development (specifically in the inner cell mass (ICM) at day 3.5 and the Epiblast at day 6.5) as well as in PGC [24].

PGC7 is recruited by *H3K9me2* and is an enzyme that prevents *Tet3* from binding to the methylated regions. This mechanism of evading demethylation through *Tet3* exclusion is shared by the maternal genome and imprinted loci on the paternal genome as *H3K9me2* is present on both sites [25]. The methylation level in the maternal genome diminishes via dilution of the DNA methylome through replication in a process called passive demethylation. *Dnmt1* helps maintain imprints on the maternal genome; only oocytes synthesize the *Dnmt1*. After fertilization of the oocyte, at the eight-cell stage, *Dnmt1* enzyme is found in higher concentrations in the nucleus. This protein can be found for as long as a week after fertilization in *human* embryos, indicating that the promoters specific to oocytes are active in the early *human* embryo or that the transcript is quite stable [26].

The demethylation of the embryo continues until the blastocyst stage, where the embryo reaches one of its lowest methylation levels (~20%) (Figure 1) [27]. However, it should be noted that complete demethylation does not take place in the preimplantation embryos. The imprinting regulatory regions, along with transposable elements, particularly intracisternal A particles (IAPs), escape this reprogramming cycle in preimplantation embryos. The sustained monoallelic expression of these genes and normal embryo development depend on the maintenance of the monoallelic methylation [17,28]. Nonetheless, there is a lack in our understanding of the maintenance of DNA methylation in such regions. It is possible that either these regions are affected by continuous demethylation and remethylation and undergo imprint stabilization quickly, as suggested by Reik and colleagues studying *mouse* embryonic stem cells (ESC) [29], or that no reprogramming takes place in these regions. Further experiments are needed to clarify which process occurs.

## 3. Tet Mutants and Knockouts

The Tet gene family is involved in embryonic development, stem cell maintenance, and lineage specification. *Tet3* is highly expressed in eggs and fertilized zygotes, but its expression decreases rapidly during cleavage, while *Tet1* and *Tet2* expression increases during pre-implantation development. Tet3 is particularly abundant in the egg cell and helps to remove DNA methylation marks from the paternal genome. If *Tet3* is absent in *mice*, the offspring do not survive beyond the perinatal period [19]. Although *Tet1* and *Tet2* have overlapping roles during embryonic development, they have distinct targets. In naive ESCs, *Tet2* is highly expressed, while in primed epiblast stem cells (EpiSC) at the start of gastrulation, it is not. Conversely, *Tet1* expression is switched on during the transition from naive to primed cells in vitro. In vivo, both *Tet1* and *Tet2* are highly expressed in the ICM before implantation, but *Tet2* expression decreases after implantation, while *Tet1* remains expressed until the post-implantation epiblast stage. From around E8.5, *Tet1* and *Tet3* are weakly expressed in the neural tube and head folds, and *Tet2* is not detected. However, by E9.5–E10.5, all three Tet genes are detected in the developing brain [30].

The research examined the contribution of *Tet1* gene to ESC maintenance and lineage specification [31]. The scientists produced *Tet1* mutant ESCs in *mice* and discovered that when the *Tet1* gene was absent, there was a partial decrease in the 5hmC levels, minor changes in global gene expression, and an imbalance in trophectoderm differentiation in vitro. Despite this, the *Tet1*-deficient ESCs remained capable of maintaining pluripotency and facilitating the development of live-born *mice* in tetraploid complementation assays. The Tet1 mutant *mice* were healthy, able to reproduce, and generally normal, but some were slightly smaller in size at birth. The findings indicate that the loss of *Tet1* has a minimal impact on pluripotency in ESCs and is suitable for normal embryonic and postnatal development [31]. However, the findings of study by Li et al., indicate that a lack of *Tet1* negatively impacts the natural capacity of *human* embryonic stem cells (hESCs) to differentiate into neuroectoderm. This is likely caused by a reduction in the expression of *PAX6*, which plays a crucial role in the formation of the *human* neuroectoderm [32].

Double-knockout (DKO) embryonic stem cells and *mice* lacking *Tet1* and *Tet2* enzymes were created in this study [31]. These DKO cells had reduced levels of 5hmC and showed defects in chimeric embryos. While some DKO embryos exhibited abnormalities and perinatal lethality, viable and normal *Tet1/Tet2*-deficient *mice* were obtained, although they had abnormal methylation at various imprinted loci. Females had smaller ovaries and reduced fertility. The study suggests that loss of both enzymes is compatible with development but can lead to hypermethylation and impaired imprinting. *Tet3* may also play a significant role in the hydroxylation of 5mC during development [33]. The *Tet1* and *Tet2* enzymes are not essential for embryonic stem cell differentiation. *Tet1* deficiency causes lethality in *mice*, and *Tet2* knockout *mice* develop leukemia later in life. Deficiency of both *Tet1* and *Tet2* leads to partial perinatal lethality, with most embryos dying by mid-gestation. Similarly, combined loss of *Tet1* and *Tet3* leads to mid-gestation lethality. The findings indicate that Tet enzymes have a compensatory role in development [34].

Although the individual loss of Tet enzymes or combined deficiency of *Tet1/2* is compatible with embryogenesis, the impact of complete loss of Tet activity and 5hmC marks is unknown. Researchers generated *Tet1/2/3* triple-knockout (TKO) ESC to investigate this. The study found that the depletion of all three Tets resulted in impaired ESC differentiation, as seen in poorly differentiated embryoid bodies and teratomas. The ESCs contributed poorly to chimeric embryos and could not support embryonic development, suggesting that Tet- and 5hmC-mediated DNA demethylation is essential for proper regulation of gene expression during ESC differentiation and development [33]. In the study by Koh et al., researchers found that *Tet1* and *Tet2* are enzymes regulated by *Oct4* that help sustain 5hmC in mESCs. When *Tet1* was depleted in ESCs using RNAi, it led to decreased expression of the Nodal antagonist *Lefty1* and resulted in hyperactive Nodal signaling and skewed differentiation into the endoderm-mesoderm lineage in embryoid bodies in vitro. *Tet1*-depleted ESCs formed hemorrhagic teratomas with increased endoderm, reduced neuroectoderm, and ectopic appearance of trophoblastic giant cells. The study suggests that 5hmC is an epigenetic modification associated with the pluripotent state, and *Tet1* functions to regulate the lineage differentiation potential of ESCs [21].

In triple-knockout (TKO) in hESCs lacking *Tet1*, *Tet2*, and *Tet3*, *Dnmt3B*-mediated hypermethylation of the Nodal signaling-related genes and *Nanog/Sox17* promoters repressed their activation and inhibited *human* PGC-like cells (hPGCLC) induction. However, knockout of *Dnmt3B* in TKO hESCs partially restored Nodal signaling and *Nanog/Sox17* expression and rescued hPGCLC induction. Tet proteins stimulated *Sox17* through the Nodal pathway and directly regulated Nanog expression at the onset of hPGCLCs induction. Therefore, the epigenetic balance of DNA methylation and demethylation in key genes plays a fundamental role in early hPGC specification [35].

## 4. De Novo DNA Methylation after the Blastocyst ICM Nadir

Once the embryo starts progressing from the blastocyst stage, and in the corresponding cultured ESC emulating naïve preimplantation pluripotency to formative post-implantation pluripotency in culture [36], the global DNA methylation level starts to increase. Two enzymes perform the de novo methylation—*Dnmt3a* and *Dnmt3b* [37]. These enzymes are responsible for the addition of a methyl group to the DNA at the five-position of cytosine. The *Dnmt3a/b* enzymes are regulated by *Dnmt3L*, an enzyme that binds to the catalytic domains of *Dnmt3a/b* and activates them [38]. Elegant structural studies of DNA-protein complexes have shown that histones with the active mark (*H3K4me3*) prevent *Dnmt3L* from binding, thereby serving as a mechanism in which histones can regulate DNA methylation at particular promoters that are turned OFF in a differentiated cell but not on promoters that remain ON in a differentiated cell [39]. *UHRF1* and *Dnmt1* work in collaboration for the maintenance of the methylation levels [26,40].

The loss of *Dnmt1* during embryonic development can result in various irregularities, such as the altered expression of imprinted genes, the activation of normally silenced IAP sequences, and the incorrect silencing of the X-chromosome. In *humans*, *Dnmt3b* mutations are associated with a recessive genetic disorder called ICF (immuno-deficiency, centromere instability, and facial anomalies) syndrome, which is characterized by immune system deficiencies, facial abnormalities, and instability of centromeres. Additionally, in *mice* lacking *Dnmt3L*, male germ cells experience reduced methylation of repetitive elements, leading to aberrant transcription during early germ cell development and decreased methylation of paternally methylated imprinted loci [5].

## 5. Maintenance of Methylation by *Dnmt1*

Methylation levels are maintained by *Dnmt1* enzymes [37], and this enzyme exists in two isoforms. The somatic isoform of *Dnmt1* undergoes alternate splicing by enzymatic cleavage to form a mature form. This isoform is expressed at low levels in oocytes and early embryos but is abundantly expressed in somatic cells. Another isoform (*Dnmt1o*) is highly expressed in early embryos and oocytes. Maintenance of parent-of-origin-specific DNA methylation on the imprints is also carried out by Dnmt1o [41].

The Dnmt1 consists of two domains: the C-terminal domain, which is similar to other Dnmts, and the N-terminal regulatory domain [42]. The nuclear localization signal (NLS), the proliferating cell nuclear antigen (PCNA) binding domain, the replication foci-targeting sequence (RFTS) domain, the CXXC domain, and two bromo-adjacent homology (BAH) domains make up the N-terminal regulatory region [42]. The *Dmap1* binding domain interacts with the histone deacetylase 2 (*Hdac2*) and the transcriptional repressor *Dmap1* [43]. *Dnmt1* is targeted to the replication foci during S-phase by the RFTS and PCNA binding domain [44,45]. RFTS occupies the catalytic domain, which could contribute to *Dnmt1*’s autoinhibitory impact [46]. These autoinhibition systems might aid in preventing abnormal DNA methylation. Only hemi-methylated CpGs can be methylated due to the CXXC domain’s ability to bind and occlude unmethylated CpGs from the active site [47]. BAH domains have also lately been linked to the localization of *Dnmt1* to replication sites, but the theoretical underpinnings are yet unknown [48].

## 6. *Tet* Enzymes Antagonistically Regulate DNA Methylation Mediated by *Dnmt* Enzymes

DNA methylation is mediated by *Dnmt* enzymes, and demethylation is mediated by *Tet* enzymes. Kyriakopoulos et al., showed how the *Tet* enzymes work antagonistically to *Dnmts* via three mechanisms: (1) by inhibiting the methylation maintenance machinery’s function in the unmethylated regions, (2) by downregulating the de novo enzymes (*Dnmt3a/b*), and (3) by ensuring efficient oxidation of 5mC at the accessible regions as *Tet* knockouts show more than 70% fully methylated CpGs as compared to only 20–30% in WT cells. *Tet* enzymes inhibit *Dnmts* activity in the highly methylated regions and protect the unmethylated regions from maintenance (*Dnmt1)* and de novo methylation (*Dnmt3a/b*), therefore ensuring that the cell-type-specific methylation patterns are formed and maintained [49].

## 7. PGC Migration during Embryonic Development

There are two ways in which PGC can be specified: inductive and determinative modes. Inductive mode, also known as epigenesis, is common in animals. In this mode, PGCs are formed from a group of embryonic cells that are influenced by signals from the surrounding cells. This ancestral way of PGC formation is seen in axolotls and mammals. On the other hand, the determinative mode, or preformation, is typical in many model organisms such as Drosophila, C. elegans, D. rerio, and Xenopus. In this mode, germ cell-specific mRNAs and proteins are synthesized during oogenesis and deposited in oocytes in the form of ribonucleoprotein complexes known as the germ plasm. Blastomeres that receive the germ plasm give rise to PGCs.

In contrast to determinative modes, organisms that rely on inductive mechanisms for PGC formation do not rely on maternally accumulated determinants in the zygote. Instead, PGCs are induced de novo from undifferentiated embryonic cells, and the specification of PGCs is orchestrated by inductive signals. *Mouse* is the best-studied model among the organisms that use inductive signaling-based mechanisms for PGC specification (Figure 1). The germ cell lineage arises during gastrulation from cells in the proximal epiblast, and cells transplanted into the proximal epiblast from elsewhere can also give rise to PGCs. Interestingly, the extraembryonic ectoderm adjacent to the proximal epiblasts expresses two members of the TGF-Beta superfamily, *BMP4*, and *BMP8*, and inactivation of either gene leads to embryos lacking PGCs. Similarly, isolated epiblasts can be induced to form PGCs by adding recombinant *BMP4* to the culture medium.

In *mice*, the E3.5–4.5 preimplantation blastocysts harbor an inner cell mass containing the initial naïve pluripotent stem cells that later give rise to all somatic cells, extraembryonic endoderm (XEN), and PGC Naïve pluripotent ESC in culture is maintained by *LIF/Jak-Stat3* with *Erk* and *Wnt/Beta-catenin* inhibitors. To induce PGC, potency factors like *Klf4/5, Rex1/Zfp42* are lost from the naïve state in the preimplantation embryo. This loss and the induction of the factors *Otx2* and *Pou3f1/Oct6* and *Dnmt3a/b* enable the formative pluripotent state/EpiLC in the post-implantation embryo at E5.5 [50]. Loss of LIF signaling gain of *FGF/Erk* signaling also enables the transition from naïve to formative pluripotency. It is only after the naïve state is downregulated that PGC arise [51]. Formative pluripotency can be induced in several mammalian species ESC and is the only period when PGC can be induced [52], and is enhanced by high *Nanog*, low FGF signaling, and low *Otx2*, in a putative small PGC precursor cell subpopulation within the formative pluripotent population. After formative pluripotency, and by the start of gastrulation at E6.5, a third primed pluripotent/epiblast stem cell (EpiSC) state arises, that is prepared to undergo gastrulation but is no longer capable of allocating PGC (Figure 1). By the end of gastrulation at E8.5, the primed pluripotent stem cells have lost *Oct4* and populated the 3 germ layers, endo-, meso-, and ectoderm, and the only remaining *Oct4*-positive, potentially totipotent cells are the PGC [53,54]. PGC are first detected after dye injection of cells in the epiblast of the E6–6.5 embryo, and within 40hr some lineage traced cells are the distinct large, alkaline phosphatase positive cells (AP+) in the posterior extraembryonic ectoderm [55]. The AP + PGC are near the extraembryonic mesoderm of the allantois, and embryonic wings of mesoderm at the posterior primitive streak and are distinct microscopically as larger cells and can be stained for alkaline phosphatase [56]. But, the allocation of PGC occurs at the formative pluripotent stage at about E5.5 before the cells become distinct, microscopically. Primed pluripotent EpiSC are sustained in culture as ESC by *Fgf4* and activin [57,58] and are no longer capable of induction to PGC [51,52,59]. In summary, the distribution of PGCs (primordial germ cell) in *mice* seems to be time-specific, occurring within a roughly 24-h period focused on formative pluripotency around E5.5, which is one day prior to the onset of gastrulation. However, our current knowledge is limited as we lack information about the markers for identifying the subpopulation of PGC precursors at E5.5. Additionally, it is possible that signaling from “undetectable” molecular events at E5.5, which become detectable later, may be necessary for community effects.

*Blimp1* plays a crucial role in the initiation of the mouse germ cell lineage as a transcriptional repressor. Its disruption causes a blockage in the early stages of PGC formation. In *Blimp1*-deficient mutant embryos, a compact group of approximately 20 PGC -like cells forms, but they do not display the typical migration, proliferation, and suppression of homeobox genes that typically accompany PGC specification. *Blimp1* is expressed extensively in stem cells of developing embryos [60]. Experiments tracing the genetic lineage show that the *Blimp1*-positive cells that arise from proximal posterior epiblast cells are the lineage-restricted precursors of PGC [61].

*Blimp1* and *Prdm14* play a critical role in epigenetic reprogramming during the specification of PGCs and early germ cells. However, *Blimp1* is not necessary for reprogramming somatic cells to iPSCs, while *Prdm14* is essential for maintaining *human* and potentially *mouse* ESCs but not *mouse* epiSCs [62]. Although *Blimp1* is not required for the derivation and maintenance of mouse ESCs or epiSCs, it can limit reversion to a pluripotent state. While *Blimp1* is needed for PGC specification, it is not necessary for *Blimp1*-deficient epiSCs to transition to rESCs, suggesting that they do not go through a *Blimp1*-positive PGC-like state. Overall, PGC specification and epiSC-to-rESC reprogramming are valuable models for studying epigenetic reprogramming mechanisms. *Blimp1* is critical for maintaining unipotent germ cells but may hinder pluripotent reversion [63,64]. *Blimp1* can directly trigger the initiation of the PGC-specific fate together with *AP2γ* and *Prdm14*. A study has determined the occupancy of essential genes by *AP2γ, Blimp1,* and *Prdm14*, uncovering a transcriptional network that relies on these three factors for PGC development. It has also been shown that *Blimp1, AP2γ,* and *Prdm14* are enough to specify PGCs, causing a unique resetting of the epigenome to a basal state [65].

Recent studies have revealed that the development of PGCs in embryos during formative stages (E5.5–E6.0) until they become large PGCs at E7.5 involves a number of important factors. These factors include *Bmp4* signaling from the extraembryonic ectoderm (ExE or XEN), which is countered by the anterior visceral endoderm (AVE), with *Bmp8b* from the ExE restricting AVE development and thus contributing to *Bmp4* signaling. Additionally, *Wnt3* in the epiblast plays a role in ensuring that the epiblast is responsive to *Bmp4.* When exposed to *Bmp4,* competent epiblast cells uniformly express key transcriptional regulators *Blimp1* and *Prdm14,* leading to the acquisition of germ-cell properties, including genome-wide epigenetic reprogramming in an organized manner [51]. Research has shown that the embryonic stem cell (ESC) state is maintained by a dynamic mechanism that involves cell-to-cell differences in sensitivity to self-renewal and susceptibility to differentiation, which are spontaneous and reversible. This condition allows ESCs to renew indefinitely and prepares them for differentiation to epiblast stem cells (EpiSCs). *Otx2*, an important transcription factor for brain development, is crucial for both ESCs and EpiSCs. It helps maintain the metastable state of ESCs by opposing ground state pluripotency and promoting differentiation commitment. *Otx2* is also necessary for the transition of ESCs to EpiSCs and for stabilizing the EpiSC state by suppressing the switch from mesendoderm to neural fate in partnership with *Bmp4* and *Fgf2* [66].

PGCs are the precursors for gametes and are one of the earliest lineages established in development [67,68]. PGC migration is a conserved process that involves chemoattraction and repulsion from different cell types [67]. In most organisms, PGC migration starts in the posterior of the embryo and is distinct from PGC proliferation, except in mammals. PGCs move passively with underlying somatic cells and actively respond to environmental cues during migration [69]. Effective migration requires cell elongation, polarity, and specific molecular pathways. The function of PGC migration is not only to allow them to reach the gonad, but also to serve as a quality-control mechanism by removing defective PGCs through negative selection [67]. The selective mechanisms may also help remove PGCs with abnormal epigenetic marks to preserve the germline [69].

PGCs, unlike other cells in the embryo, undergo maturation during migration through the developing embryo towards their destination, the gonad. During this migration period, there are significant changes in the epigenetic marks of PGCs compared to the surrounding somatic cells. These changes include reduced levels of methylated cytosine and increased levels of certain types of methylated histones [70,71]. These epigenetic marks play a role in establishing the unique epigenetic profile of PGCs typically associated with gene repression. However, the exact significance of these methylation differences for PGC function is still not fully understood.

Another crucial epigenetic event in PGC biology involves changes in X-inactive specific transcript (Xist) RNA, which are specific to female embryos and occur on the X chromosomes. In female *mice* with two X chromosomes, Xist RNA coats one of the X chromosomes to maintain its transcriptional repression. This mechanism ensures balanced gene expression between males and females. However, research led by de Napoles and Sugimoto found that during migration, PGCs in female embryos experience a reduction in Xist RNA levels. This reduction occurs exclusively in PGCs and not in somatic cells. The PGC-specific reduction of Xist RNA is crucial as germ cells need to maintain two active X chromosomes for proper gamete formation [72,73]. During meiosis, the homologous X chromosomes undergo DNA exchange, requiring two functionally equivalent X chromosomes.

## 8. Epigenetic Reprogramming and Random X-Chromosome Inactivation (XCI)

The paternal X chromosome undergoes silencing at the two-cell embryo stage, which marks the occurrence of XCI in female *mice*. Some distinguishing features of the silenced X-chromosome (Xi) are being wrapped with *Xist* and accumulating *H3K27me3* and its correlated enzymes-*Eed* (embryonic endoderm development) and *Ezh2* (enhancer of Zeste homolog 2) [74,75,76,77,78]. *Eed* and *Ezh2* are part of the *PRC2* (Polycomb repressor complex 2), which methylates Lys-9 and Lys-27 of histone H3, leading to transcriptional repression of the affected target genes. This repressed state of the paternal X-chromosome is maintained until the formation of the blastocyst. After that, the inactivated paternal X chromosome is reactivated within the following 15 h by the ICM (inner cell mass) cells that will create the epiblast [75]. The epiblast cells lack the distinctive features of the inactivated X-chromosome during this brief phase, which corresponds to the time right before implantation in *mice*. After that, the repressive markers of the inactivated X chromosome are once again re-established [wrapping by *Xist,* aggregation of *H3K27me3*, mediated by activated enzymes *Eed* and *Ezh2* of the *PRC2;* however, in the ICM, X-chromosome inactivation is random on either the paternal or maternal chromosome rather than just the paternal X chromosome (random XCI) [79].

The long non-coding RNA (lncRNA) *Xist*, which is extensively produced and associates with the inactive X chromosome in cis, has been demonstrated to be the primary regulator of XCI initiation in *mice* (Xi). A chromosomal-wide coating results from the *Xist* transcript’s *cis* distribution over the X chromosome [80,81,82,83]. Several additional chromosome-wide epigenetic modifications are driven by *Xist* aggregation on the Xi. *Ezh2* transforms *H3K27me2* to *H3K27me3* in the Xi by first attracting *PRC2* (Polycomb repressive complex 2) complexes, which comprise the enzymes *Ezh2* and *Eed.* Additionally, the Xi exhibits high DNA methylation, the addition of non-canonical histones such as macro-H2A, a lack of the activating *H3K4* methylation and *H3K9* acetylation, and a rise in *H3K9me2*. The recruitment of *PRC2* to the X chromosome through *H3K27me2* marks is thought to be important for the stabilization and spreading of *Xist* RNA, as well as the establishment of stable heterochromatin and the silencing of genes on the inactive X chromosome. Therefore, the functional significance of *H3K27me2* recruiting *Ezh2* and the *PRC2* complex is to establish a repressive chromatin state on the X chromosome that facilitates the initiation and maintenance of XCI, ensuring proper gene dosage compensation in female cells. These alterations ensure long-term transcriptional silence by strongly condensing the chromatin into a perinuclear structure known as the Female-specific Barr body [83,84,85]. Barr bodies are a result of X-chromosome inactivation (XCI), which is a process that occurs during early embryonic development in female mammals to balance the dosage of X-linked genes with males that only have one X chromosome. XCI occurs in two waves: the first wave is imprinted and occurs in the early embryo, while the second wave occurs in the developing female germline. In the naïve state, female ESCs have two active X chromosomes and do not form Barr bodies. During the transition from naïve to primed state, female ESCs may undergo random XCI, resulting in the formation of Barr bodies in a subset of cells. However, this XCI is often incomplete and reversible. In the formative and primed states, female ESCs can undergo XCI upon differentiation into specific cell types, resulting in the formation of Barr bodies.

Over a thousand genes are silenced due to *Xist* RNA aggregation along the Xi, in addition to a series of chromatin changes [86]. Specific regions in the *Xist* RNA have particular roles. The A-repeat region is responsible for inducing the silencing of genes by recruitment of *SPEN* protein which allows repressive complexes such as Histone deacetylases *(HDAC) 3* and nuclear receptor corepressor (*NCOR*)/silencing mediator of retinoic acid and thyroid hormone receptor (*SMRT)* to interact with each other [87,88,89]. The Xist RNA binding protein recruits the Polycomb-group repressive complex (*PRC*) through its interaction with the B and C repeat regions. [90,91,92]. In fact, after Xist RNA coating, Xi enriches quickly with *PRC1*-dependent *H2AK119Ub* and then with *PRC2*-dependent *H3K27me3* [89,93,94,95]. Heterogeneous Nuclear Ribonucleoprotein K (*hnRNPK)* is recruited directly by Xist RNA, which binds non-canonical *PRC1,* enabling the quick deposition of *H2AK119Ub* [90,91]. *PRC2* co-factors may recognize *H2AK119Ub*, allowing *H3K27me3* to be deposited de novo [96]. It is important to note that Polycomb protein complex accumulation is not essential for the start of gene silencing, as evidenced by the ability of Xist mutants missing the B and C repeat domain to still cause XCI, albeit with significantly reduced effectiveness [90,92]. Conversely, *PRC2* allows for stable repression to be maintained, specifically in the case of extra-embryonic lineages [97]. Although the dynamics of polycomb group (PcG) mark accumulation have been examined using chromatin immunoprecipitation-sequencing (ChIP-seq), it is still not known how precisely this relates to the *Xist* RNA coating of the Xi. In fact, the time between the overexpression of *Xist* RNA and the deposition of *H3K27me3* is unknown [98].

## 9. Understanding Genomic Imprinting by ICR and Imprinted DMRs in Epigenetics

The *H19* gene locus is an area of interest when studying genomic imprinting, a process where DNA methylation markings are established in a paternal or maternal specific way on one allele of either a gene or control element of a gene. This causes monoallelic expression, which is regulated by the imprinting control region (ICR). Germline DMRs (gDMR) refer to the ICR in the germline that has acquired monoallelic DNA methylation and serves as primary imprinting marks. Global demethylation post-fertilization and pre-implantation or de novo methylation post-implantation has no effect on the imprinted DMRs. These DMRs are different between the two gametes, and trans-acting factors can distinguish these imprinted DMRs from all the other CpG islands in the genome [99]. In the mouse genome, 28 imprinted DMRs derived from germline are known, and these preserve the methylation in their alleles throughout their lifespan. In *humans*, 38 such imprinted DMRs are known. Few studies have shown transient imprinted domains in *mice*; they survive the demethylation post-fertilization but become methylated on the paternal allele after the implantation [1,100]. This is a simplified version of the regulation of ICR; the *H19* gene locus is more complex than initially thought, and while it is known that some of the *H19* gene’s functions are performed through miR-675, there are likely other functions that do not involve this miRNA. Both the *H19* lncRNA and miR-675 are up-regulated in many types of cancers and by common triggers, but it is unclear how this occurs since miR-675 is processed at the expense of *H19* lncRNA. It is possible that miR-675 has its own regulatory sequence that has not been investigated. Additionally, the interplay between the various products processed from the *H19* gene locus remains largely unknown [101].

Congenital imprinting disorders (IDs) are characterized by changes in imprinted chromosomal regions and genes, which are responsible for parent-of-origin specific gene expression. The understanding of IDs has significantly expanded in recent years, encompassing various alterations in regulation, dosage, or DNA sequence that disrupt imprinted gene expression. This broad spectrum of changes leads to diverse clinical syndromes. However, underlying these diverse conditions are shared molecular mechanisms and common clinical impacts on growth, development, and metabolism. Consequently, the comprehensive analysis of IDs not only reveals fundamental principles of molecular epigenetics in health and disease but also facilitates personalized diagnosis and management for affected individuals and families [102]. An approach by Pham et al., enabled the generation of iPSC lines with balanced methylation for control purposes and iPSC lines from ID patients with unbalanced methylation. *Human* iPSCs thus represent a valuable cellular model for investigating the underlying mechanisms of IDs and assessing potential therapeutic strategies in relevant tissues [103].

## 10. Exploring the Potential of ICRs for Studying Windows of Early Exposures to Toxicants and their Effects on Genomic Imprinting

It has long been recommended to employ subtle changes in DNA methylation marks as exposure proxies, and the methylation patterns of ICRs offer a unique “epigenetic responsive” window to early exposures to toxicants [104]. ICRs are special in that DNA methylation marks are established early, there is commonality between different cell types and tissues, and there is possible long-term stability that propagates an appropriate short-term, transcriptional regulatory, stress response into inappropriate long-term response. These qualities enable their widespread usage as durable archives of early developmental exposures that may affect adult metabolic and other developmental processes, as well as behavioral processes [105,106]. As a result, ICRs are sensible targets for examining the earliest stages of disease utilizing readily available cell types acquired at various ages. To recognize early exposures, a preset reference panel of ICRs may be essential [105].

The Dutch Hunger Winter study of 1976 provided evidence for a relationship between the environment and the epigenome, specifically in relation to famine [107]. The study found that children of pregnant women exposed to famine, during the last year of World War II in Holland, early in gestation, were more susceptible to chronic metabolic diseases in adulthood, possibly due to persistent epigenetic differences, including changes in DNA methylation patterns and genomic imprinting [108]. The *Igf2* and *H19* genes are important in this process, as their mis-regulation can have consequences for fetal growth [109]. Studies have suggested that alterations in genomic imprinting may link abnormal fetal growth to disease susceptibility later in life, driven by maternal exposures, and this theory is supported by the Dutch Hunger Winter study [107].

## 11. Endocrine-Disrupting Chemicals and Their Potential Effects on Epigenetic Reprogramming and Transgenerational Inheritance

*Humans* are constantly exposed to endocrine-disrupting chemicals EDCs through various routes, and these chemicals have non-monotonic dose responses, meaning low-dose effects cannot be predicted by high-dose effects seen in toxicological studies [110]. This makes it difficult to identify the mechanisms by which EDCs cause adverse health effects, especially when exposures occur during critical developmental periods [111]. Many EDCs have estrogenic and/or hormonal properties, but their potential to modify epigenetic reprogramming events, such as genomic imprinting, has not been fully explored. The epigenome is particularly vulnerable to environmental exposures during early development, when dynamic changes in DNA methylation are needed for normal tissue development [112]. Recent studies in animals and *humans* have been carried out to investigate the effects of EDC exposures on imprinted genes during vulnerable developmental periods such as periconceptional, gestational, and early postnatal development, these studies have been summarized and reviewed by Bartolomei et al. [113].

A review by Tahiliani et al., describes a study on transgenerational inheritance of induced epigenetic variation in *mice*. The researchers found that supplementing the diet of pregnant female *mice* with methyl donors induces CpG hypermethylation at the agouti viable yellow (A(vy)) allele in A(vy)/a offspring, and this epigenetic inheritance occurs at A(vy) when passed through the female germline. However, they found that there was no cumulative effect of diet-induced A(vy) hypermethylation across successive generations, suggesting that this type of epigenetic inheritance occurs in the absence of additional epigenetic modifications that normally confer transgenerational epigenetic inheritance at the locus [114]. Resenfeld et al., showed that maternal exposure to bisphenol A and genistein has minimal impact on the coat color of Avy/a offspring but increases the likelihood of giving birth to agouti *mice* rather than nonagouti *mice* [115]. In a study by Sallan et al. (2009), it was observed that male offspring exposed to bisphenol A (BPA) during the perinatal period exhibited long-lasting effects on their reproductive health. Specifically, these males showed lower sperm counts and reduced sperm motility. Notably, these phenotypic changes persisted across multiple generations up to the F3 population [116].

## 12. DNA Demethylation in PGCs after Allocation from Formative Pluripotent Epiblasts: Second Wave

The development of PGCs is conserved across mammals [117]. Epigenetic reprogramming occurs in the PGCs after they migrate to the genital ridge and before they are differentiated in a sex-specific manner [118,119]. A few studies using the immunofluorescence staining technique have shown that around E8.0 in *mice*, the *H3K9me2* (repressive histone modification) becomes depleted globally [120,121]; *H3K27me3* levels gradually increase globally and at pericentric heterochromatin, *H3K9me3* levels are high [120,121,122]. From day E8.5, DNA methylation levels start declining in the epiblast from 70% to almost 4% by E13.5 in the PGCs. Global demethylation is achieved through a passive mechanism in the PGCs. *Dnmt3a/b* are down-regulated along with *Uhrf1* (recruiting factor for Dnmt1) by *Prdm1* and *Prdm14*. Down-regulation of both de novo and maintenance methylation facilitate replication-coupled DNA demethylation when the proliferation of PGCs takes place [121,123,124,125,126,127]. Several loci show the presence of DNA methylation during this pre-gonadal PGCs phase, and Dnmt1 conditional deletion results in the decline of methylation levels in almost all genomic elements, including endogenous retrovirus-intracisternal A particle, meiotic genes, and ICR [128].

Regulation of the level of DNA methylation in various genomic regions is not entirely understood; however, this DNA methylation is restored in the males at E13.5 and in the females post-birth. Some studies show that enzymatic conversion of 5mC to 5hmC plays a vital role in demethylation, particularly for imprints. The imprints are protected from global demethylation until E9.5–10.5, when the PGCs migrate to the genital ridge [129]. The mechanism of Tet-mediated demethylation is explained in chapter 3. In PGCs, at E9.5-E 11.5, a global increase in *5hmC* and a decrease in *5mC* indicate upregulation of *Tet1* and *Tet2* enzymes [129,130]. While knockout studies in *mice* demonstrate that germline DNA demethylation can occur independently of *Tet1* or *Tet2* [33,131,132]. *Tet1* and *Tet2,* are necessary for the demethylation of meiotic gene promoters and effective imprint erasure for germ cell development [132].

## 13. Consequences of Altered Epigenetic Reprogramming in PGCs for Germ Cell Development, Reproductive Health, and Offspring Health

Altered epigenetic reprogramming in PGCs has been suggested to have significant consequences for germ cell development, reproductive health, and offspring health. The disruption of normal epigenetic processes during PGC reprogramming can lead to long-lasting effects on the germ cell lineage and subsequent generations [133].

One potential consequence of altered epigenetic reprogramming in PGCs is the aberrant establishment or erasure of DNA methylation marks. DNA methylation patterns play crucial roles in gene regulation and genomic stability, and any deviations from the normal reprogramming process may result in aberrant gene expression patterns and epigenetic instability in germ cells. These alterations have the potential to disrupt germ cell development, compromise gamete quality, and contribute to reproductive disorders, including infertility and an increased risk of miscarriages.

Moreover, altered epigenetic reprogramming in PGCs can impact reproductive health by influencing the susceptibility to epigenetic modifications induced by environmental factors. PGCs are highly sensitive to their developmental microenvironment, and disruptions in their epigenetic reprogramming may alter their response to environmental cues during development. This altered sensitivity can render germ cells more susceptible to epigenetic modifications induced by various factors such as endocrine disruptors, stress, diet, and lifestyle choices. These epigenetic changes can persist in subsequent generations, potentially affecting the reproductive health and fertility of offspring. Furthermore, altered epigenetic reprogramming in PGCs may have implications for offspring health and disease susceptibility (reviewed in detail in [5]).

Epigenetic marks acquired during PGC reprogramming can influence gene expression patterns in somatic cells and contribute to the establishment of cellular identity. Disruptions in this process can lead to altered gene expression profiles in offspring, which may underlie an increased risk of various diseases, including metabolic disorders, neurodevelopmental disorders, and cancer. Additionally, intergenerational transmission of epigenetic alterations resulting from disrupted PGC reprogramming has been proposed as a potential mechanism for the inheritance of certain traits and disease susceptibility across generations (reviewed in [134]).

It is important to note that the consequences of altered epigenetic reprogramming in PGCs are complex and multifaceted, and further research is needed to fully elucidate the specific mechanisms and outcomes. The study of animal models and *human* cohorts with perturbed PGC reprogramming, as well as advanced genomic and epigenomic techniques, holds promise for unraveling the precise consequences of disrupted epigenetic reprogramming and its implications for germ cell development, reproductive health, and offspring health.

## 14. Non-CpG Methylation (CpA, CpT and CpC) and Its Unique Role in Epigenetic Reprogramming

Non-CpG methylation, involving the methylation of cytosine residues in DNA sequences other than CpG dinucleotides, has emerged as a fascinating area of research in the field of epigenetic reprogramming. While CpG methylation has traditionally received significant attention due to its well-established role in gene regulation and genomic stability, recent studies have highlighted the distinct features and functional implications of non-CpG methylation, including CpA, CpT, and CpC methylation. Initially considered a rare phenomenon, non-CpG methylation has been observed in various contexts, such as embryonic stem cells, neuronal development, and disease states. Importantly, non-CpG methylation exhibits dynamic patterns during development and cellular differentiation, suggesting its involvement in epigenetic reprogramming processes. Non-CpG methylation is predominantly scarce in adult somatic cells, representing merely 0.02% of the total 5mCs observed in such cells. Conversely, non-CpG methylation patterns exhibit variations among *human* pluripotent cell types. Notably, *human* male embryonic stem (ES) cells, specifically the H1 cell line, demonstrate substantial methylation levels of approximately 25% at non-CpG sites [135,136]. In contrast, *human* female ES cells, such as the H9 cell line, exhibit lower levels of methylation compared to H1 ES cells at both CpG and non-CpG sites [135,136,137]. This discrepancy may be attributed to reduced expression levels of DNMTs in female ES cells, leading to diminished de novo methylation [137]. In both *mice* and *humans*, non-CpG methylation is observed throughout the genome in adult brain tissue [100,138,139,140]. Interestingly, different types of brain cells display distinct levels of non-CpG methylation. Unlike CpG methylation, which is predominantly maintained through the activity of DNA methyltransferase 1 (DNMT1), non-CpG methylation is resistant to DNMT1-mediated maintenance. Instead, non-CpG methylation is believed to be established and maintained by de novo DNA methyltransferases, such as DNMT3A and DNMT3B [8,141]. This distinctive mechanism suggests that non-CpG methylation may serve as a robust and stable epigenetic mark that contributes to long-term gene regulation and cellular identity.

The functional implications of non-CpG methylation in epigenetic reprogramming are still being elucidated. Additionally, aberrant non-CpG methylation patterns have been observed in various diseases, including cancer and neurodevelopmental disorders, underscoring its potential as a diagnostic and therapeutic target [142,143]. Understanding the mechanisms underlying non-CpG methylation establishment, maintenance, and functional consequences is an active area of research. Various factors, including DNA sequence context, chromatin accessibility, and specific DNA methyltransferases, likely contribute to the establishment and regulation of non-CpG methylation patterns. Investigating the crosstalk between non-CpG and CpG methylation, as well as their interplay with other epigenetic modifications, will provide further insights into the complexity of epigenetic reprogramming processes.

## 15. Histone Modifications during Epigenetic Reprogramming

Gene activation is typically linked to hypomethylation at the promoter regions. How is gene expression suppressed during global demethylation in *human* PGC development? Histone covalent modifications have crucial functions in controlling gene expression and embryonic development and serve as essential bearers of epigenetic information. The correlation between gene expression and distinct histone modification marks has been identified by genome-wide studies in cell lineages. Generally, methylation on histone *H3K9* and *H3K27* is closely correlated with repressive regions, while histone H3 lysine 4(*H3K4*) methylation is typically linked with permissive promoters and enhancers. According to recent findings, both in the mouse and *human* species, early PGC development involves a reorganization of chromatin modification globally. *H3K9me3* is mostly preserved at pericentric heterochromatin during PGC development, and repressive *H3K27* trimethylation (*H3K27me3*) is enriched throughout this process. This could lead to the regulation of gene expression in the globally hypomethylated PGC genomes [121,144]. In mouse PGCs, these markers have also been linked to the suppression of retrotransposons. The accessibility of important regulatory DNA regions, in addition to DNA methylation and histone modifications, is directly related to the transcriptional regulation of gene expression. So, to spatiotemporally control the gene expression, chromatin accessibility needs to be established appropriately [145]. The histone modification during epigenetic reprogramming has been extensively reviewed in [146,147,148].

## 16. Placental Epigenetics: How the Imprinting and X-Inactivation Differs from Embryo

ESC and trophoblast stem cells (TSC) are distinguishable after five cell divisions. *Cdx2* becomes dominant in outside cells and *Oct4* is dominant in inside cells [149]. TSC and ESC diverge at 5th division (approximately 32 cell stage but cell divisions are not synchronous in mammals, so this is approximate). Extra-embryonic endoderm (XEN) arise from ESC soon after this six to seven cell divisions and by 8th cell division late naïve pluripotent cells can’t make XEN. XEN are also paternal X inactive like TSC, so ESC must return to plastic random X-inactivation after XEN has delaminated and ESC no longer make XEN (probably 7th to 8th cell division). The imprinting and X-inactivation occurs in the ESC and TSC not only after they diverge, random ESC inactivation also must occur after XEN has been set aside by ESC in the late preimplantation blastocyst [150].

The question of whether the maternal or paternal X chromosome is preferentially inactivated has also generated interest. The paternal X is selectively inactivated in extraembryonic tissues in *mice* [151], possibly as a result of different *XIST* methylation in sperm compared to eggs [152]. Furthermore, the rat yolk sac preferentially inactivates the paternal X [153]. In *mice*, some extraembryonic membranes and extraembryonic lineages that eventually give rise to the placenta have paternally imprinted XCI [154,155,156]. Rats, cows, and marsupial mammals have all been reported to have XCI that is maternally imprinted. The embryonic lineages in *mice* that eventually give rise to the rest of the fetus, however, have random XCI [154]. Mule and horse placenta have also been found to contain random XCI [157].

This study describes confined placental mosaic (CPM), a condition where mosaicism exists solely within the placenta, affecting about 2% of viable pregnancies [158]. Unlike generalized mosaicism, which is identified by the presence of two or more cell lines with different chromosome configurations in both the fetus and its placenta, CPM refers to the occurrence of chromosomal mosaicism that affects only the placental tissue, not the fetus. It is a condition where the cytogenetic abnormality, usually trisomy, is limited to the placenta. CPM is detected by chorionic villus sampling during pregnancy and is typically caused by mitotic chromosomal nondisjunction, resulting in aneuploidy. CPM aneuploidy can cause complications including increased incidence of preterm births, low birth weights, intrauterine growth restriction, and intrauterine death of a chromosomally normal fetus. CPM is surprising given that the placenta and embryo differentiate from the outer cell mass and inner cell mass, respectively, following zygote formation [159,160].

## 17. Concluding Remarks

In this review, epigenetic reprogramming and the developmental toxicology in *mice* embryos from fertilization to the development of PGC are discussed. Despite advances in techniques, there are several questions that remain to be answered. The similarities and differences between *mice* and *human* epigenetic reprogramming is summarized in Table 1.

The underlying mechanism of how *Tet2* promotes both DNA and RNA demethylation, and how it interacts with the CpG islands, is not fully understood. Some studies have indicated that *Tet2* requires Zscan proteins (*Zscan4f*) for regulating cellular processes. *Zscan4 SCAN* domain proteins (SCAN domains come from the mammalian retroviruses) bind to domains that are present on *Tet1*, *2*, *3*, so in addition to *Tet2*, the Scan proteins might also bring all three *Tets* to the DNA [161]. *Zscan* proteins are not detected in lower organisms, which could mean that given the large structure of this protein, other domains might be facilitating the binding of *Tet* enzymes to the CpG islands or that recruitment of *Tets* to the methylation sites differs between lower and higher organisms. Furthermore, *Zscan* is only present in the 2CE-like stage in higher organisms and not detected in PGC. The role of *Zscan4* and other *SCAN* proteins in the regulation of *Tet* enzymes and choosing targets is not yet fully understood. It is unknown how the PGCs remethylate the imprinted loci and the retrotransposons, including *LINE-1* and *IAP* elements. The PGCs are very few, and that adds to the complexity of studying them. Testicular cancer is known to be caused by the activation of the *c-kit* gene, which encodes a receptor known as *Kit*. The normal function of this receptor is to bind to the Steel ligand and sustain the proliferation of PGCs during development. However, in cases of testicular cancer, the activation of c-kit can cause PGCs to persist into adulthood without undergoing normal maturation processes. Interestingly, studies have shown that these PGCs have zero DNA methylation, which may contribute to their abnormal behavior and ultimately lead to the development of testicular cancer [162]. The role of histone modifications in controlling the DNA de novo methylation and demethylation also remains to be investigated. Understanding the interaction of epigenetic, genetic, and environmental factors during the reprogramming events remains vital for our understanding of several diseases.

## 18. Future Directions

In the field of epigenetics, there are several areas that require further investigation to gain a better understanding of the mechanisms underlying DNA methylation and demethylation, and how they impact fertility. One such area is the role of histone modifications in controlling DNA de novo methylation and demethylation, which could have significant implications for the development of eggs and sperm. The mechanisms by which *Tet* enzymes carry out both DNA and RNA demethylation, as well as the role of *Zscan4* and other *SCAN* proteins in regulating *Tet* enzymes and selecting targets, are also not yet fully understood and may have implications for fertility. Additionally, research into the process by which PGCs remethylate imprinted loci and on the retrotransposons, including *LINE-1* and *IAP* elements, is important to better understand how these mechanisms may impact fertility. While the scarcity of PGCs makes this difficult to study, studies into testicular cancer caused by activated cKit have revealed that PGCs can be present in adults with zero DNA methylation. It is important to determine which animal models are most appropriate for drawing accurate conclusions about DNA methylation in *humans*. Another area that requires further exploration is the mechanism by which PGCs completely demethylate, even in the presence of *Dnmt3a* and *Dnmt3b.* The role of *Dnmt1* in de novo DNA methylation remains a topic of debate, with some studies suggesting that it only occurs in vitro and not in vivo. Investigating how these mechanisms impact fertility and the development of eggs and sperm is critical for understanding their clinical significance. Although epigenetic changes are potentially reversible, it is still important to continue research in this area to develop epigenetic treatments that involve CRISPR. Understanding the mechanisms underlying DNA methylation and demethylation could help to develop more effective treatments for diseases such as cancer and other genetic disorders and may have significant implications for fertility and reproductive health.

**Table 1 cells-12-01874-t001:** Table showing the differences and similarities between the epigenetic reprogramming in *mice* and *humans*.

Aspect	*Mice*	*Humans*
DNA Methylation	DNA methylation undergoes erasure during PGC development [163].	DNA methylation is partially retained in mature gametes [10]
Histone-to-Protamine Transition	Histones are replaced by protamines during sperm maturation [164]	Histones are replaced by protamines during sperm maturation [164]
DNA Methylation Dynamics in Spermatogenesis	Re-establishment of DNA methylation during spermatogenesis [165]	Mechanisms of DNA methylation re-establishment not well explored
DNA Methylation Patterns in Oocytes	Unique bimodal pattern, predominantly in gene bodies (~40% methylation in oocytes) [1]	Higher average DNA methylation, predominantly in gene bodies (~54% methylation in oocytes) [166]
DNMT3L Expression	Essential for de-novo methylation in mouse oocytes [167]	Not expressed in *human* oocytes [168]
Retained Histones in Mature Sperm	Few nucleosomes are retained in mature sperm [169]	More nucleosomes are retained in mature sperm [170].
DNA Methylation	Paternal protamines replaced by maternal histones, erasure of almost all paternal DNA methylation. Maternal DNA methylation largely preserved.	Global reprogramming of DNA methylation in the pre-implantation embryo, with substantial retention of maternal methylation. Less passive demethylation, possibly due to a more active role of DNMT1 [139].
Zygotic Genome Activation (ZGA)	Major wave of ZGA at the 2-cell stage [171].	Major wave of ZGA at the 8-cell stage [172].
Chromatin Remodeling	Relaxed chromatin state in zygotes gradually resolved to a more canonical state by the blastocyst stage [173].	Widespread open chromatin in pre-ZGA embryos, rapidly remodeled upon ZGA. Temporal regulation of chromatin accessibility dependent on transcriptional activation [174].
Transcriptome	Differences in the transcriptome compared to *humans*. Similar transcription factors, but divergent regulation and networks [175].	Similar transcription factors, but temporal regulation and networks can differ [175].
De novo Methylation	Two phases of de novo methylation: first in the paternal genome in the zygote, second between the 4- and 8-cell stage coinciding with ZGA. Transient methylation of repeat elements [176].	De novo methylation observed during pre-implantation development. Two phases: early-to-mid pronuclear stage in the paternal genome, and between the 4- and 8-cell stage coinciding with ZGA. Methylation of repeat elements, transient in subsequent developmental stages [177].
Function of Repressive Chromatin	Repressive chromatin marks such as *H3K27me3* play a role in reinforcing lineage specification in both *mice* and *humans* [178].	The targeted gain of *H3K27me3* is observed in the post-implantation embryo in *mice*, but the specific mechanisms and extent of *H3K27me3*’s function in *humans* are largely unexplored [178].
Role of H3K9me2	*H3K9me2* is associated with methylated DNA in the post-implantation embryo in both *mice* and *humans*. However, its functional role is specialized and not required for the genome-wide gain of DNA methylation [179].	*H3K9me2*’s specific functions in the post-implantation embryo in *humans* are not well understood, and its role may be different from *mice*.
Active Chromatin Marks	Active histone modifications like *H3K4me3* and *H3K27ac* likely play a role in transcriptional regulation during lineage specification in both *mice* and *humans* [178].	The specific requirements and effects of *H3K4me3* and *H3K27ac* in establishing and reinforcing the transcriptional program during lineage specification may vary between *mice* and *humans* [178].
Imprinted Gene Clusters	Conserved in methylation status, allelic expression, and synteny [17]	Conserved in methylation status, allelic expression, and synteny, with several exceptions [17]
Number of Imprinted Genes	~151 [180]	50–90 [181]
Identification Methods	Sequencing approaches over SNPs, genomic imbalances [181]	Sequencing approaches over SNPs, genomic imbalances [181]
Regulatory Complexity	Imprinted gene expression and methylation may be more widespread and variable [180]	Imprinted gene expression and methylation may be more widespread and variable than *mice* [182]
Maintenance of Imprints	Requires ZFP57 and other genetic factors during later stages [183]	Maintenance of imprints during *human* reprogramming is not well understood [178]
Placental Imprinting	Limited number of placental-specific imprinted gDMRs [184]	More than 1500 placental-specific imprinted gDMRs, mostly not conserved between species [185]
Imprinting mechanisms	DNA methylation, *H3K27me3* (extra-embryonic lineages) [184]	DNA methylation, *H3K27me3* (unknown if present)

## Figures and Tables

**Figure 1 cells-12-01874-f001:**
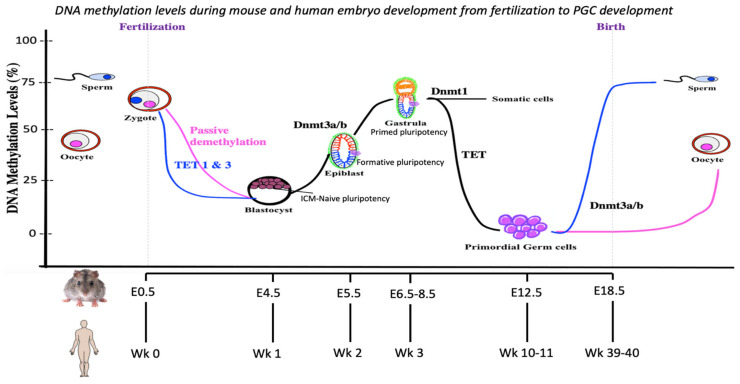
DNA methylation levels during *mouse* and *human* embryo development from fertilization to PGC development. The methylation levels of the CpG islands are shown in the figure. The sperm genome is highly methylated (~75–80%), and the oocytes are relatively less methylated (~50%). After fertilization, global demethylation mediated by Tet enzymes occurs in the parental genomes, dropping the methylation to ~20% at the blastocyst stage. The process is rapid in the paternal genome and gradual in the maternal genome. The ICR remain methylated during this demethylation. The methylation is then restored gradually from the blastocyst stage to the gastrula stage by *Dnmt3a* and *Dnmt3b* enzymes. The inner cell mass (ICM) in the blastocyst is in naïve pluripotent stage, the epiblast comprises formative pluripotent state and beginning of gastrulation is primed pluripotency state before the cells divide into endoderm, mesoderm, and ectoderm by the end of gastrulation. PGCs first appear at E5.5 in the formative pluripotent stage and become prominent by E6.5. *Dnmt1* maintains the DNA methylation levels in somatic cells. The second round of global demethylation occurs again in the PGC by the action of Tet enzymes. The PGCs have the least methylated genome in the entire lifespan of the *mice* as the parent-of-origin imprints are erased (~3–4%) [10] and new ones are established. The methylation of the sperm gradually increases by the action of *Dnmt3a* and *Dnmt3b* and is restored before birth. However, in oocytes, the methylation remains low and is restored at puberty. The timings for epigenetic events for both *mice* and *humans* are shown at the bottom of the figure.

## Data Availability

Not applicable.

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
