# Peer review of "Epigenetic Reprogramming in Mice and Humans: From Fertilization to Primordial Germ Cell Development"

_cells, 2023, doi:10.3390/cells12141874_

Round 1

Reviewer 1 Report

Review Cells, MDPI:

Epigenetic Reprogramming in mouse and humans: from fertilization to primordial germ cell development

Authors: A. Singh, D.A. Rappolee, D.M. Ruden

May, 2023

The review entitled “Epigenetic Reprogramming in mouse and humans: from fertilization to primordial germ cell development” written by Singh et al. deals with epigenetic reprogramming events, starting post-fertilization in the early embryo and continuing throughout primordial germ cell development in both sexes. The main focus is on waves of DNA methylation, catalyzed by TET enzymes and DNA methyltransferases, and their connection to developmental potency. Within this scope, embryonic X-inactivation mechanisms and genomic imprinting are discussed and a description of various methods to study DNA methylation is provided.

Major comments:

The authors of this very extensive review try to cover many interesting and important topics that deal with epigenetic reprogramming in primordial germ cells (waves of DNA (de)methylation, responsible enzymes) and a possible impact on reproduction and offspring health (putative inter- and transgenerational consequences), in case these reprogramming events are disturbed (EDC).

Unfortunately, the authors get lost in details that have been already extensively handled in other reviews (e.g. describing signaling events and transcription factors in PGC specification and migration). However, the most important topics are only briefly described and not always discussed in the context of the current literature (the impact of environmental effects or EDCs on PGC epigenetic reprogramming and putative health consequences) although this is announced in the abstract.

All in all, the review should be streamlined and it would be of great interest to discuss putative consequences of altered epigenetic reprogramming in PGCs for germ cell development, reproductive health, offspring health (inter-, transgenerational inheritance).

Line 34: is “clean state” a recognized definition?

Line 38/39: DDM1 and 2 are genes found in plants, e.g. Arabidopsis thaliana, maize; not in mammals / mice or humans (in mammals DNA demethylation is catalyzed by TET1,2,3 (tet methylcytosine dioxygenase 1,2,3))

Lines 45/46 […] some repeat-poor (Satellite DNA, Alu elements, LINE-1 elements) regions, […]

Wrong context; examples for repetitive DNA elements should not be directly given when “repeat-poor” is explained  might lead to misunderstanding

Lines 43 -53

-   Line 52: "[…] which my help to allow their expression and transposition." What do the authors mean?

-   Retroelements / transposable elements are regulated in the germline by different epigenetic, transcriptional and posttranscriptional mechanisms to also protect the germ line and to maintain genomic integrity - please discuss this more balanced / carefully.

            (e.g. Zeng and Chen, Genes, 2019; Fukuda, Elife, 2022; Wang eet al., Cell Regen 2020)

-   Ref 4: Guo et al., Cell. 2015

These authors describe that "This indicates that, during the global DNA demethylation process in the PGCs, the evolutionarily younger and more active transposable elements maintain higher levels of residual methylation than the older and less active ones. […] This pattern indicates that the evolutionarily younger transposable elements are transcribed more actively than the evolutionarily older ones. In addition, the evolutionarily younger and more active repeat elements, such as L1, Alu, and ERVK (Figures 7 and S5B), had relatively high levels of residual DNA methylation, even when the whole genome of the male PGCs in the 11 week embryos and the female PGCs in the 10 week embryos were nearly devoid of any methylation.

Figure 1:

-   Figure is rather common and found in a similar depiction in other manuscripts.

-   DNA methylation: please specify CpG islands or global DNA methylation?

-   Include time points of embryonic development for better understanding.

Figure 2:

-       Figure is rather common and found in a similar depiction in other manuscripts.

-       Upper panel (maternal site): arrow should point from enhancer to the ICR and not vice versa

Chapter 7: PGC migration during embryonic development

-       Very detailed chapter on PGC specification and migration in various species, including signaling cascades, transcription factors and developmental potency; insights into epigenetic events and mechanisms are hardly given

-       There are already many excellent reviews on PGC migration and specification.

Chapter 9: Understanding genomic imprinting by ICR and imprinted DMRs in epigenetics

-       Short passage, though very interesting topic; more studies could be included and discussed.

Chapter 11: “Endocrine disrupting chemicals and their potential effects on epigenetic reprogramming and transgenerational inheritance”

-       Very interesting and important topic, which should be discussed in context of putative effects e.g. on the epigenome in PGC / reprogramming events in PGCs and consequences for reproductive potency and offspring health. Unfortunately, there is only a very short description of what EDC are and that they might influence changes in DNA methylation pattern.   

-       Moreover, the famous agouti viable yellow mouse model is mentioned here, which serves as an epigenetic sensor in response to (diet-induced) epigenetic alterations. But where is a (possible) connection to EDC?

Chapter 12: DNA demethylation in PGCs after allocation from formative pluripotent epiblasts: Second wave

-       Line 503 – 513: the role of TET enzymes and 5mC and 5hmC can already be found in chapter 3; could one connect these chapters on first and second wave in order to avoid a repetition?

Chapter 13: Histone modifications during epigenetic reprogramming

-       The main focus of this review is on DNA methylation. The chapter on posttranslational modifications is too short and does not cover the role of histone PTMs during PGC development. Although announced in the abstract, it might be better to skip and instead refer to other excellent reviews.

Chapter 15: Techniques for detection of epigenetic changes in DNA/RNA

-       RNA methylation has not been discussed in the review. Why is it described here?

Use of references:

-       Many reviews cited, esp. in chapter 1

-       Over 80 references older than 10 years (published 2013 and before 2013)

Grammar / Spelling / Formatting

-       Line 44: primarily FOUND in the proximal promoter

-       Explanation of abbreviations is not uniform (either more than once (e.g. Xi, Xist, ), or there is no explanation (e.g. line 399 hnRNPK; line 190 hPGCLC), or not explained when the abbreviation was used the first time, but only later on and multiple times (e.g. line 136 ESCs and embryonic stem cells (ESC) in line 144, 180, 329; triple knockout line 173, 188 (TKO))

-       Missing words: e.g. line 397: Xist RNA binding protein

-       Noun in “plural”, corresponding verb in “singular”

-       Missing punctation marks

Reviewer 2 Report

The review brings already consolidated concepts, in a clear writing, well organized sequence of events.

Some minor points could be improved:

- When stating the percentage of methylation during reprogramming, more citations are needed (pre and after fertilization, differences between males and females, etc).

- 4. "De novo DNA remethylation" - this sentence is confusing - "De novo" seems the proper term, and therefore, "remethylation" should be replaced by "methylation".

- Figure 2 does not bring any new data. Consider deleting or improving this figure.

- Item 15: RNA should be deleted. The item refers to DNA methylation.

Overall, the review could greatly benefit from more recent information on specific topics, some examples are: IVG as an in vitro model for other studies; or epigenetic regulation using CRISPR; KO models for gametogenesis, and all of them.

Reviewer 3 Report

Please find below my comments, suggestions and questions.

In the beginning of their review the authors should briefly introduce some basic aspects of DNA methylation at DNA CpG islands/shores (and mention non-CpG methylation as well), its functional role, and list all enzymes executing methylation and demethylation in mice and humans.

Every time while speaking about mice-specific or human-specific epigenetic events or mechanisms the authors should clearly state it, or even separate parts of the texts into the corresponding sub-chapters. Now, in many cases, unless the reader checks the quoted article it’s hard to figure out whether the authors talk about mice or human. Thus, I find the review rather poorly structured, even though it is divided into several sub-chapters. The review is entitled “Epigenetic reprogramming in mouse and humans’, but the authors tend to focus on epigenetic of mice, they even state in the very first sentence of the abstract - “In this review, advances in the understanding of epigenetic reprogramming from fertilization to the development of primordial germline cells in a mouse embryo are discussed”.

What about Non-CpG methylation (CpA, CpT and CpC) and its unique role in epigenetic reprogramming? What about chromatin organisation changes, mentioned in the abstract and rather omitted in the text? Do the authors want to describe changes in 5hmC/5mC ratio during the embryogenesis and beyond?

The schematic illustration, comparing epigenetic events in mice and humans, their similarities and differences, would be a good addition to the text. Also, it might help to guide other researchers when deciding whether mice model is an appropriate model to use in their experiments, given the aforementioned differences.

Speaking about methylation of repetitive elements of genome, the authors may want to discuss methylation of ribosomal DNA (rDNA), methylation of telomere regions, etc. What about DNA methylation of Human endogenous retroviruses (HERVs)?

Line 10: “Understanding the molecular basis of many diseases require knowledge” – requires knowledge?

Line 41. “DNA maintenance methylation machinery [3]”. - would not it be better to say “DNA methylation maintenance machinery” instead?

Round 2

Reviewer 1 Report

The revised review has significantly improved. All important points have been addressed. Further interesting sections and points to discuss have been included, underlining the importance of epigenetic mechanisms in developing PGCs.

However, the review is very long though. If it is not possible to further streamline the review, one could consider removing the chapter on "Techniques to detect / edit DNA methylation" and publish it as a separate manuscript.

ok, only few spelling mistakes

Reviewer 3 Report

Please find below two comments regarding the updated version of the text.

1) I suggest removing this sentence (Line 36). Summary statements about what is novel in this  review: from my perspective it is that toxicant and stress stimuli that affect the unique period of allocation at formative pluripotency may affect quantity and quality of PGC, and what is new from a mechanistic side on new insights into normal and abnormal epigenetics that affect health?

2) Figure 1. Please take a closer look, there are red wavy underlines highlighting Wk abbreviation. It's clearly a Word formatting error, please rectify.

Overall, in the revised form the review is a high-quality work and will benefit research community.
